



# Computing Extreme Storm Surges in Europe Using Neural Networks

Tim H.J. Hermans[1], Chiheb Ben Hammouda[2], Simon Treu[3], Timothy Tiggeloven[4], Anaïs Couasnon[4,5], Julius J.M. Busecke[6], and Roderik S.W. van de Wal[1,7]

[1]Institute for Marine and Atmospheric Research Utrecht, Utrecht University, Utrecht, The Netherlands
[2]Mathematical Institute, Utrecht University, 3584 CD Utrecht, The Netherlands
[3]Transformation Pathways, Potsdam Institute for Climate Impact Research (PIK), Member of the Leibniz Association, 14412 Potsdam, Germany
[4]Institute for Environmental Studies, Vrije Universiteit Amsterdam, 1081HV Amsterdam, the Netherlands
[5]Deltares, Delft, The Netherlands
[6]Lamont-Doherty Earth Observatory, Columbia University, Palisades, NY, USA
[7]Department of Physical Geography, Utrecht University, Utrecht, 3584 CB, The Netherlands

**Correspondence:** Tim H.J. Hermans (t.h.j.hermans@uu.nl)

**Abstract.** Because of the computational costs of computing storm surges with hydrodynamic models, projections of changes in extreme storm surges are often based on small ensembles of climate model simulations. This may be resolved by using data-driven storm-surge models instead, which are computationally much cheaper to apply than hydrodynamic models. However, the potential performance of data-driven models at predicting extreme storm surges is unclear because previous studies did

not train their models to specifically predict the extremes, which are underrepresented in observations. Here, we investigate the performance of neural networks at predicting extreme storm surges at 9 tide-gauge stations in Europe when trained with a cost-sensitive learning approach based on the density of the observed storm surges. We find that density-based weighting improves both the error and timing of predictions of exceedances of the 99th percentile made with Long-Short-Term-Memory (LSTM) models, with the optimal degree of weighting depending on the location. At most locations, the performance of the

neural networks also improves by exploiting spatiotemporal patterns in the input data with a convolutional LSTM (ConvLSTM) layer. The neural networks generally outperform an existing multi-linear regression model, and at the majority of locations, the performance of especially the ConvLSTM models approximates that of the hydrodynamic Global Tide and Surge Model. While the neural networks still predominantly underestimate the highest extreme storm surges, we conclude that addressing the imbalance in the training data through density-based weighting helps to improve the performance of neural networks at

predicting the extremes and forms a step forward towards their use for climate projections.



## 1 Introduction

Through strong winds and low atmospheric pressure, storms can cause abnormally high coastal water levels called storm surges. In Europe and elsewhere, storm surges have led to numerous coastal floods, some resulting in many casualties and substantial socioeconomic losses (Paprotny et al., 2018). Due to climate change, the frequency and height of extreme sea levels

are expected to increase globally, primarily due to sea-level rise (Hermans et al., 2023; Jevrejeva et al., 2023; Vousdoukas et al., 2018). Although likely to a smaller extent, extreme sea levels may also change due to changes in atmospheric conditions driving storm surges (Muis et al., 2020; Vousdoukas et al., 2018; Muis et al., 2023; Shimura et al., 2022). However, projections of atmospherically driven changes in extreme storm surges are typically based on small ensembles of climate model simulations. Consequently, the uncertainties of these projections due to differences between climate models and internal climate variability

are large (Muis et al., 2023; Hermans et al., 2024)

An important reason why projections of extreme storm surges are often based on only a few climate model simulations is that global climate models do not simulate storm surges reliably, if at all. Instead, the atmospheric changes simulated by climate models need to be translated to changes in storm surges with another model. Typically, computationally expensive, high-resolution hydrodynamic models are used for this (e.g. Muis et al., 2020; Vousdoukas et al., 2018; Muis et al., 2023; Shimura

et al., 2022). However, data-driven storm-surge models based on regression, gradient boosting, neural networks and other machine learning techniques are emerging (see Qin et al., 2023, for a review) that, once trained, may be used as computationally cheaper alternatives to hydrodynamic models to translate climate model simulations to changes in storm surges.

So far, data-driven storm-surge models have primarily been used to predict short time series of local water levels or peak heights during specific events, using the characteristics of tropical cyclones traveling over the region as predictors (Ayyad et al.,

2022; Lockwood et al., 2022; Ramos-Valle et al., 2021; Ian et al., 2023; Sun and Pan, 2023; Naeini and Snaiki, 2024, among others). Other studies have applied data-driven models to gridded atmospheric reanalysis data to reconstruct continuous time series of storm surges (Tausia et al., 2023; Cid et al., 2018, 2017; Tiggeloven et al., 2021; Bruneau et al., 2020; Tadesse et al., 2020; Tadesse and Wahl, 2021; Harter et al., 2024). In principle, these reconstructions can then be used for extreme value analysis (Cid et al., 2018; Tiggeloven et al., 2021). However, previous studies did not specifically train their models to predict

the extremes.

Compared to more moderate storm surges, extreme storm surges are underrepresented in the training data. Without addressing this data imbalance during training, data-driven models may be biased toward more common events (Krawczyk, 2016). This could explain why existing data-driven models typically underestimate extreme storm surges (e.g., Tadesse et al., 2020; Tiggeloven et al., 2021; Harter et al., 2024), although limitations of the input data and the selection of predictor variables also

play a role (Harter et al., 2024). Therefore, the potential performance of data-driven models at predicting extreme storm surges is still unclear. Furthermore, because several previous studies did not focus on evaluating the extremes and/or considered extremes exceeding relatively low thresholds (e.g., Bruneau et al., 2020; Tadesse et al., 2020; Tiggeloven et al., 2021), it is also not fully clear how neural networks compare to state-of-the-art hydrodynamic models.



A second hurdle toward using data-driven models to project changes in extreme storm surges is their application to climate
model simulations, which are typically provided at a lower resolution than the atmospheric reanalyses used by previous studies.
For instance, the climate-model simulations from the High Resolution Model Intercomparison Project (Haarsma et al., 2016)
that were used by Muis et al. (2023) to force their Global Tide and Surge Model (GTSM) have a spatial resolution comparable
to the ERA5 atmospheric reanalysis (Hersbach et al., 2020), but are provided at a temporal frequency of 3 hours at best. The
simulations of other models participating in the Coupled Model Intercomparison Project 6 (Eyring et al., 2016) are typically
also provided at a relatively low temporal resolution. Optimal model architectures and hyperparameter combinations that were
found using hourly or more frequent observational training data may therefore not apply in the context of projecting changes
in extreme storm surge.

In this study, we investigate how well neural networks can compute extreme storm surges based on atmospheric reanalysis
data when the imbalance of moderate v.s. extreme storm surges is addressed during model training. To address the imbalance,
we use the cost-sensitive learning approach *DenseLoss* (Steininger et al., 2021) that weights the contribution of prediction errors
to the training loss according to the rarity of their target observations, derived with kernel density estimation. Additionally, we
trained the neural networks with 3-hourly observational data because of the underlying aim to eventually apply them to climate
model simulations.

We analyzed 9 tide-gauge locations in western Europe, which are all subject to mainly extratropical cyclones, but vary in
their oceanographic setting. We show how the performance of the neural networks at predicting extreme storm surges at these
locations depends on how much additional weight rare data points are given, and whether the neural networks are designed to
exploit only temporal or also spatiotemporal patterns in the input data. Additionally, we compare the performance of neural
networks trained with and without density-based weighting to that of the multi-linear regression (MLR) model of Tadesse et al.
(2020) and the hydrodynamic model GTSM (Muis et al., 2020, 2023).

## 2   Methodology

### 2.1   Data preparation

We trained neural networks to predict storm surges at 9 tide-gauge locations in western Europe, selected based on data availability and geographical coverage (see Figure 1). We limited our analysis to 1979-2017 because for that period, GTSM simulations
are available for comparison (Muis et al., 2020). As predictands, we used hourly, quality-controlled tide-gauge observations
from the GESLA3 database (Haigh et al., 2021). To derive non-tidal residuals (hereon referred to as storm surges) from the
tide-gauge observations, we first subtracted annual means and then tides predicted through harmonic analysis performed with
the T-Tide MATLAB package (Pawlowicz et al., 2002). Following Tadesse et al. (2020), we used 67 tidal constituents for the
harmonic analysis, including only years of observations with at least 75% data availability. While we verified the tidal amplitudes and phases that we estimated by comparing them with the estimates of Piccioni et al. (2019), the tide predictions are not
perfect and residual tidal signals may remain despite our correction (Tiggeloven et al., 2021). However, to avoid smoothing out
high-frequency surge variability, we did not attempt to remove potential residual tidal signals further by low-pass filtering.



As predictors, we used hourly data from the atmospheric reanalysis ERA5 (Hersbach et al., 2020). The explanatory variables we used are zonal, meridional and absolute wind speed at 10m above the surface and atmospheric pressure at sea level, in a box of 5 by 5 degrees (20 by 20 grid cells) around each tide gauge (Figure 1). As shown by Figure 1, the storm surge at time $t$ was predicted using predictors at time $t$ and up to 24 hours prior. We did not test other time windows.

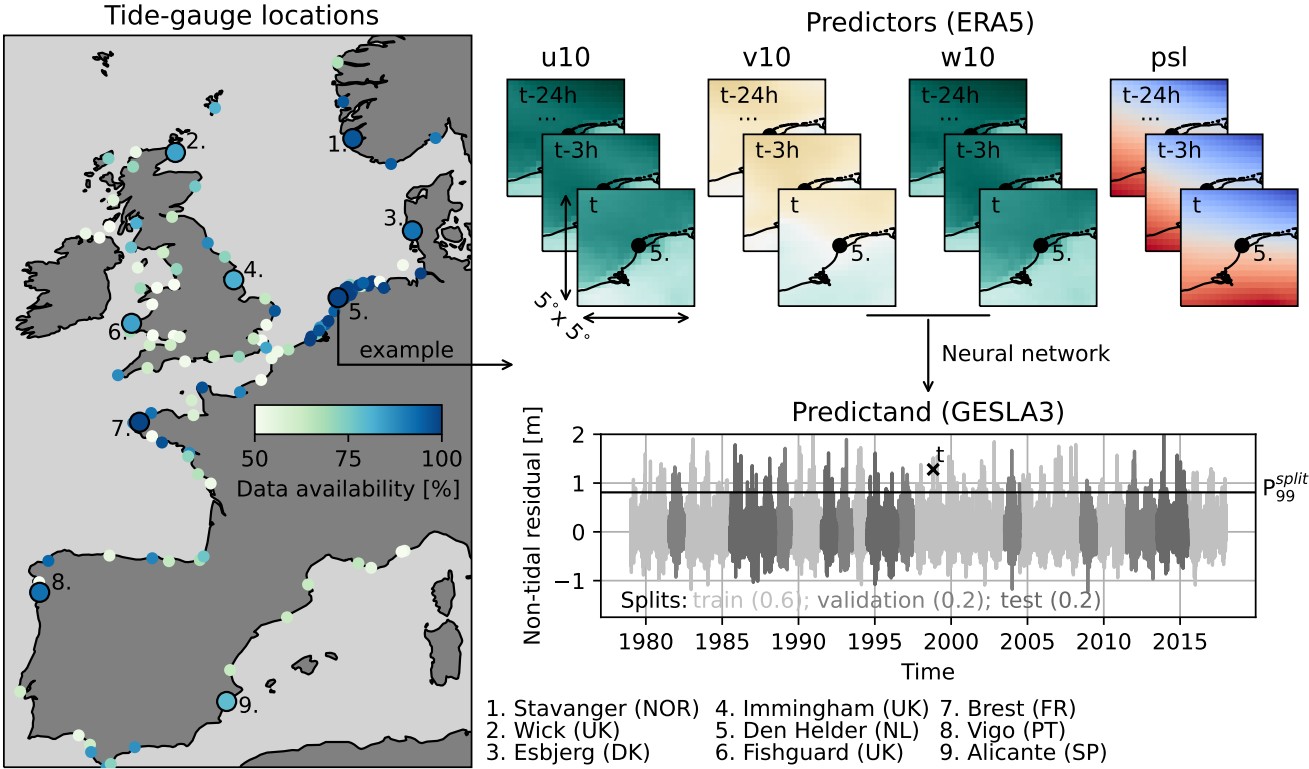

**Fig. 1.** Data availability (percentage of 3-hourly time steps with processable observations during 1979-2017) at European tide gauges with at least 20 years of observations, the 9 selected tide-gauge locations, and an example of the prediction of a non-tidal residual (referred to as storm surge) at at an arbitrary time step $t$ at location 5 (Den Helder, NL). The abbreviations $u10$, $v10$, and $w10$ stand for zonal, meridional and absolute wind speed at 10m above the surface, respectively, and $psl$ for the atmospheric pressure at sea level. The different colors of grey indicate how the full time series of storm surges is divided up into splits, and $P_{99}^{split}$ denotes the 99th percentile of all observed storm surges in a given split.

Before training and evaluating the neural networks, we subtracted the annual means and mean seasonal cycle from both the predictand and predictor variables at each time step to avoid these signals from dominating what the models will learn. Additionally, we subsampled the hourly predictand and predictor data every 3 hours to mimic the highest temporal frequency at which climate model simulations are typically provided (see Section 1). The time series were then split into non-overlapping train, validation and test portions containing 60%, 20% and 20% of the available data, respectively (see Figure 1). We used the



train split to train the models, the validation split to evaluate training convergence and tune the hyperparameters of the models (further explained in Section 2.3), and the test split to evaluate model generalization to unseen data.

Because we observed that at some locations, chronological splitting led to a particularly uneven distribution of extreme storm surges over the different splits, we applied a simple stratified sampling scheme. This involved splitting the timeseries

into years from July to June, stratifying the years based on the height of the 99th percentile of storm surges ($P_{99}$) in each year, and randomly assigning years from each stratum to the splits according to the aforementioned split-size ratios. The randomness of the stratification was controlled with a seed. Finally, both the predictand and predictor data in each split were standardized by subtracting the mean and dividing by the standard deviation of the data in the train split. The predictions obtained with the standardized predictor data were back-transformed accordingly before evaluation.

## 2.2 LSTM and ConvLSTM models

For each location, we tested two neural network architectures: one with a long short-term memory (LSTM) layer and one with a convolutional LSTM (ConvLSTM) layer, both followed by three densely connected layers (see Figure A1). An LSTM layer is a type of recurrent neural network that can capture temporal dependencies in sequential data (Hochreiter and Schmidhuber, 1997). A ConvLSTM layer is an LSTM layer in which internal operations are convolutional (Shi et al., 2015), and can therefore

also capture spatiotemporal dependencies. The choice for an LSTM model is motivated by the results of Tiggeloven et al. (2021), who found that LSTM models generally predict storm surges better than basic artificial neural networks, convolutional neural networks and ConvLSTM models. Since Tiggeloven et al. (2021) used predictors in a region of only 1.25 by 1.25 degrees around each tide gauge, we additionally used the ConvLSTM model to test whether an LSTM model also outperforms a ConvLSTM model when using predictors in a 5 by 5 degree region. To develop the models, we largely followed the designs of

Tiggeloven et al. (2021) but made a few tweaks that we found were beneficial for either model performance or efficiency. Both models were developed with the python-package TensorFlow (TensorFlowDevelopers, 2024). The software that we developed to train and evaluate the models is publicly available (Hermans, 2025). Further details and flowcharts of the model architectures are provided in Appendix A.

## 2.3 Model training & hyperparameter tuning

Following previous studies (e.g., Bruneau et al., 2020; Tiggeloven et al., 2021), we trained our models separately at each tide gauge, with the commonly used mean square error (MSE) loss function. The MSE loss minimizes the mean of the squared differences between all predictions and observations, and while it therefore penalizes larger errors more than smaller errors, it does not directly address the underrepresentation of extremes in the training data. Therefore, we implemented the cost-sensitive learning approach DenseLoss (Steininger et al., 2021) by multiplying each squared error between a prediction and

an observation by a weight inversely proportional to the density of the observation obtained through kernel density estimation. The density-based weights are given by $f_w$:





$$f_w(\alpha, y) = \frac{max(1 - \alpha p(y), \epsilon)}{\frac{1}{N}\sum_{i=1}^{N} max(1 - \alpha p(y_i), \epsilon)}, \tag{1}$$

in which $y$ is the observation, $p(y)$ is the normalized density function of the observations, $\epsilon$ is a positive, real-valued constant ($10^{-6}$) that clips the weights of observations with the highest densities to a non-zero value, and $\alpha$ is a hyperparameter controlling the strength of the density-based weighting (the higher $\alpha$, the stronger the weighting). Here, we test $\alpha$ values of 0 (no weighting), 1, 3 and 5. Figure B1 shows an example of the corresponding density-based weights of standardized observations in the train split at Den Helder (NL). We refer to (Steininger et al., 2021) for further details of the DenseLoss method.

Like Bruneau et al. (2020), Tiggeloven et al. (2021) and Harter et al. (2024), we trained our models using Adam optimization (Kingma and Ba, 2017). We used a maximum of 100 training epochs and stopped training if the loss of the validation split did not further decrease during 10 consecutive epochs. The network weights corresponding to the epoch with the lowest validation loss were stored. Based on preliminary tests, we used a batch size of 128 time steps. Due to the dimensions of the training data, extensive tuning of all hyperparameters was too computationally costly, especially for the ConvLSTM model. Therefore, we kept most hyperparameters of the model architectures defined in Appendix A constant. However, because we observed that the learning and dropout rates had a relatively strong influence on the evolution of the training and validation loss, we varied the learning rate between $1e^{-5}$, $5e^{-5}$ and $1e^{-4}$ and the dropout rate between 0.1 and 0.2. Combined with the 4 different values of $\alpha$, this resulted in 24 unique sets of hyperparameters. To save computation time, we trained the LSTM models with all 24 settings, but the ConvLSTM model only with $\alpha$=5, informed by the results for the LSTM models (see Section 3.1). Additionally, to account for variance in the results due to randomness in the initialization and optimization of model weights, we trained the models 5 times with each set of hyperparameters.

## 2.4 Performance evaluation

As discussed in Section 2.1, we evaluated the LSTM and ConvLSTM models using their predictions in the validation and test splits. For each split, we computed the root mean square error (RMSE) between the predictions and observations, considering only the time steps in that split at which the observed storm surges are extreme. We defined extremes using the 99th percentile in each split ($P_{99}^{split}$) as a threshold. Additionally, we only included exceedances if they were part of an event consisting of at least two exceedances within a time span of 12 hours. This was done to avoid including exceedances potentially primarily arising from dominant semi-diurnal tidal signals that may not have been fully removed with the harmonic analysis explained in Section 2.1. We refer to the resulting root mean square error as $RMSE_{P99}^{split}$.

The $RMSE_{P99}^{split}$ conveys the error of predictions of observed extremes in a split regardless of whether the predictions are extreme. As falsely predicted extremes would also be included in an extreme value analysis of the predictions, we additionally evaluated whether extremes are predicted at the right time. To do so, we used $P_{99}^{split}$ as a threshold to count the number of false positive (#FPs), false negative (#FNs), true positive (#TPs) and true negative (#TNs) predictions in each split. We then computed the corresponding F1 score, which ranges from 0 to 1, by taking the harmonic mean of the precision ($=\frac{\#TPs}{\#TPs+\#FPs}$) and recall ($=\frac{\#TPs}{\#TPs+\#FNs}$):



$$F1_{P99}^{split} = \frac{2 * precision_{P99}^{split} * recall_{P99}^{split}}{precision_{P99}^{split} + recall_{P99}^{split}}. \tag{2}$$

To place the performance of the neural networks in perspective, we also computed the error metrics introduced above for the predictions of the MLR model of Tadesse et al. (2020), which uses empirical orthogonal functions (EOFs) of gridded sea-level pressure and zonal and meridional near-surface winds as predictors. We trained the MLR model with the same ERA5 data used to train the neural networks except that we had to reduce the size of the predictor data from 5 by 5 to 4.5 by 4.5 degrees around each tide gauge to manage the dimension constraints of the principal-component analysis of SciPy. This may lead to a

moderately lower performance (Tadesse and Wahl, 2021).

Additionally, we compared the performance of the data-driven models to that of GTSM (Muis et al., 2020, 2023); which is a global, state-of-the-art hydrodynamic model. We derived the error metrics for GTSM from the surge component of GTSMv3.0 simulations forced with sea-level pressure and surface winds from ERA5. We used the simulations of Muis et al. (2020) instead of Muis et al. (2023) because these consistently agreed better with the observations. While we trained the data-driven models

with the same atmospheric data, GTSM was forced with hourly instead of a 3-hourly ERA5 data. GTSM simulations forced with 3-hourly ERA5 data are unavailable, but would likely have a larger error relative to observations (Agulles et al., 2024).

## 3    Performance of the neural networks

### 3.1    Effect of density-based weighting on LSTM models

Figure 2 shows the RMSE of the predictions of extreme observations versus the F1 score in the validation split ($\text{RMSE}_{P99}^{val}$

versus $\text{F1}_{P99}^{val}$), for each tide-gauge location. Each circle denotes these error metrics for an individual LSTM model. The $\text{RMSE}_{P99}^{val}$ is displayed relative to the height of the 99th percentile ($P_{99}^{val}$) at each location so that it can directly be compared between locations. Depending on the location, the minimum $\text{RMSE}_{P99}^{val}$ among the LSTM models trained without density-based weighting ($\alpha$=0, dark blue) ranges from 0.29 to 0.62 times $P_{99}^{val}$. Immingham stands out as a location at which the LSTM models have a relatively high $\text{RMSE}_{P99}^{val}$. Density-based weighting clearly reduces the $\text{RMSE}_{P99}^{val}$ of the LSTM models at all

locations (Figure 2). Furthermore, LSTM models trained with a higher $\alpha$ value (higher degree of weighting) tend to have a lower $\text{RMSE}_{P99}^{val}$, as seen by the gradient of dark blue circles ($\alpha$=0) on the right to red circles ($\alpha$=5) on the left of each plot.

While among all tested values $\alpha$=5 leads to the lowest $\text{RMSE}_{P99}^{val}$ at each location, the improvement obtained through density-based weighting differs between locations (e.g., compare Stavanger to Den Helder). On average, increasing $\alpha$ from 0 to 5 reduces the minimum $\text{RMSE}_{P99}^{val}$ by 28%. The minimum $\text{RMSE}_{P99}^{val}$ for $\alpha$=5 ranges from 0.22 to 0.35 times $P_{99}^{val}$ at all

locations except Immingham, where the minimum $\text{RMSE}_{99}^{val}$ exceeds 0.45 times $P_{99}^{val}$. As seen by the bars at the bottom of each panel in Figure 2, the spread in the $\text{RMSE}_{P99}^{val}$ among LSTM models with the same $\alpha$ is typically between 0.07-0.15 times $P_{99}^{val}$, depending on the location and on $\alpha$.

Density-based weighting also influences the $\text{F1}_{P99}^{val}$ score of the LSTMs models. The maximum $\text{F1}_{P99}^{val}$ score of models trained without density-based weighting ($\alpha$=0) ranges from 0.28 at Alicante to 0.70 at Den Helder (Figure 2). Increasing $\alpha$





from 0 (dark blue) to 1 (light blue) improves the median $F1^{val}_{P99}$ at all locations and the maximum $F1^{val}_{P99}$ somewhat at most locations. However, increasing $\alpha$ further has a mixed effect. For instance, at Esbjerg and Alicante, training with $\alpha=5$ leads to the highest $F1^{val}_{P99}$, while at several other locations, $\alpha=1$ leads to the highest $F1^{val}_{P99}$ (Figure 2).

     In general, the effect of density-based weighting on $F1^{val}_{P99}$ is moderate: on average, the maximum $F1^{val}_{P99}$ score of LSTM models trained with density-based weighting ($\alpha=1$, 3 or 5) is 9% higher than the maximum $F1^{val}_{P99}$ score obtained without

density-based weighting ($\alpha=0$). The effect of $\alpha$ on the F1 score results from the partial compensation between the precision and the recall of the LSTM models, which depend on $\alpha$ oppositely (see Figure C1). Namely, increasing the density-based weights generally leads to more true positives but also to more false positives, similar to the forecaster's dilemma (Lerch et al., 2017). While Figure 2 suggests that increasing $\alpha$ beyond 5 may reduce the $RMSE^{val}_{P99}$ even further, this will therefore likely lead to less precise LSTM models, with a lower $F1^{val}_{P99}$.

In conclusion, density-based weighting improves the overall performance of the LSTM models at predicting extremes at all locations, at least with an $\alpha$ value of 1. However, the optimal $\alpha$ value depends on both the location and the metric to optimize. Finally, we note that while density-based weighting may improve the performance of the LSTM models at predicting extremes, it reduces their performance at predicting moderate observations by construction. Depending on the intended application of the neural networks, this may also need to be considered.





**Fig. 2.** Scatter plots of the root mean square error of predictions of the extreme storm surges observed in the validation split, relative to the 99th percentile of all observations in the validation split ($\frac{1}{P_{99}^{val}} \times \mathrm{RMSE}_{P99}^{val}$) [-], versus the F1 score of predictions in the validation split evaluated using $P_{99}^{val}$ ($\mathrm{F1}_{P99}^{val}$) [-], for each tide-gauge location. Each circle denotes these error metrics for an individual LSTM model. The colors indicate the different values of $\alpha$ (0, 1, 3 or 5) used to train each LSTM model (30 LSTM models per $\alpha$ per location, as explained in Section 2.3). The bars on the bottom and left sides of each panel denote the minimum, median and maximum relative $\mathrm{RMSE}_{P99}^{val}$ and $\mathrm{F1}_{P99}^{val}$ of the LSTM models for each $\alpha$, respectively. The value of $P_{99}^{val}$ is shown in the upper right corners.




## 3.2 Comparison between models

Next, we compare the performance of the LSTM models with that of (1) the ConvLSTM model, (2) the MLR model of Tadesse et al. (2020), and (3) the hydrodynamic model GTSM (Muis et al., 2020). To this end, Figure 3 shows the same error metrics as Figure 2, but only for the best LSTM models for each $\alpha$ (colored circles). These were selected based on the highest sum of the model's rankings for $\text{RMSE}_{P99}^{val}$ and $\text{F1}_{P99}^{val}$, separately for each location. The best ConvLSTM models were selected in the same way (trained only with $\alpha$=5; black-edged red squares).

First, we find that the selected LSTM models have a higher $\text{RMSE}_{P99}^{val}$ than the selected ConvLSTM models at all locations except at Vigo and Alicante (Figure 3a), and a lower $\text{F1}_{P99}^{val}$ at all locations except Alicante (Figure 3b), at least for $\alpha$=5. Using a ConvLSTM layer instead of an LSTM layer leads to an average improvement in the $\text{RMSE}_{P99}^{val}$ and $\text{F1}_{P99}^{val}$ of 6 and 13%, respectively. The finding that the ConvLSTM model outperforms the LSTM model at most locations indicates that exploiting spatiotemporal patterns in the input data is generally beneficial for predicting extreme storm surges.

Second, Figure 3a shows that at all locations, the LSTM models trained with density-based weights have a lower $\text{RMSE}_{P99}^{val}$ than the MLR model of Tadesse et al. (2020) (white triangles). The LSTM models trained without density-based weights ($\alpha$=0; dark blue circles) also have an $\text{RMSE}_{P99}^{val}$ similar to or lower than the MLR model, except at Vigo. Furthermore, the $\text{F1}_{P99}^{val}$ score of the LSTM models is higher than that of the MLR model at all locations, regardless of $\alpha$ (Figure 3b). These results suggest that the non-linear relations between extreme storm surges and atmospheric predictors that the LSTM models capture, but the MLR model cannot, are important to consider. The difference in the performance between the LSTM- and MLR models may partially be reduced by using wind stress instead of wind speed as a predictor because wind stress is related to surge more linearly (Harter et al., 2024). Like for the LSTM models, the performance of the MLR model may also be improved by incorporating density-based weighting in its optimization, but we did not test this.

Third, the selected LSTM models have a higher $\text{RMSE}_{P99}^{val}$ than the hydrodynamic model GTSM (white diamonds) at all locations except at Wick, Vigo and Alicante, and a lower $\text{F1}_{P99}^{val}$ at all locations except Brest and Vigo, regardless of $\alpha$ (Figures 3a & b). On average, GTSM has an approximately 12% lower $\text{RMSE}_{P99}^{val}$ than the minimum $\text{RMSE}_{P99}^{val}$ and a 12% higher $\text{F1}_{P99}^{val}$ than the maximum $\text{F1}_{P99}^{val}$ of the LSTM models. Hence, we conclude that at the majority of locations, a state-of-the-art numerical model like GTSM outperforms relatively simple neural networks like the LSTM models, although the hourly instead of 3-hourly atmospheric forcing that was used to drive GTSM (see Section 2.4) may also help. Especially at Immingham, GTSM performs relatively well while the LSTM models perform relatively poorly (Figure 3), indicating that the LSTM models have poorly learned the existing relationship between the predictors and the extreme storm surges at that location.

Finally, since the ConvLSTM models outperformed the LSTM models at most locations, we find that the ConvLSTM models perform more similarly to GTSM than the LSTM models (Figure 3). Except at Stavanger and Immingham, the performance of the ConvLSTM models trained with $\alpha$=5 closely approaches or even exceeds that of GTSM. The average relative differences in the $\text{RMSE}_{P99}^{val}$ and $\text{F1}_{P99}^{val}$ between the best ConvLSTM models and GTSM at these 7 locations are marginal. Hence, based on these evaluation metrics, the ConvLSTM model may be a viable alternative to state-of-the-art hydrodynamic models.





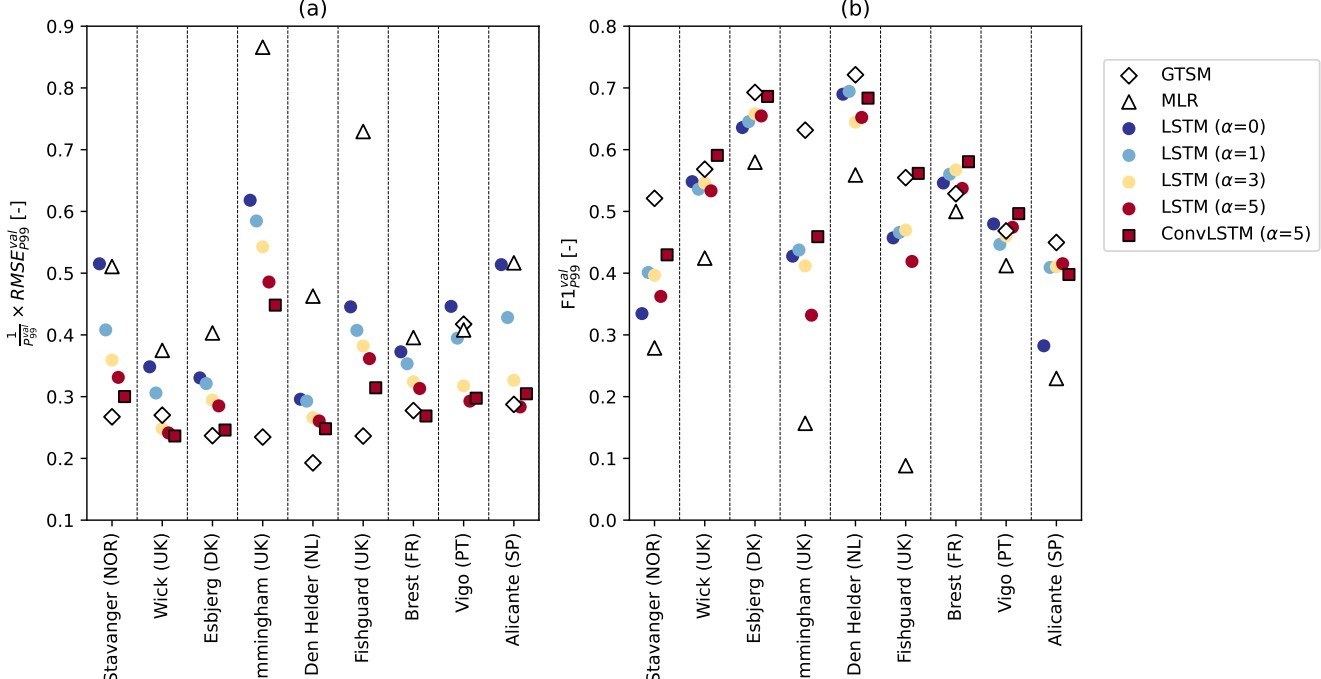

**Fig. 3. (a)** $\mathrm{RMSE}_{P99}^{val}$ relative to $\mathrm{P}_{99}^{val}$ [-] and **(b)** $\mathrm{F1}_{P99}^{val}$ [-], for the best overall performing LSTM models for each $\alpha$ (colored circles), the best overall performing ConvLSTM model for $\alpha$=5 (black-edged red squares), the MLR model of Tadesse et al. (2020) (white triangles) and the hydrodynamic model GTSM (Muis et al., 2020) (white diamonds), at every tide-gauge location.

### 3.3 Model generalization

So far, we only considered how well the different models perform in the validation split. In this section, we also evaluate how well the models generalize. To do so, we compare the error metrics in the validation and test splits (Figure 4), the latter of which was completely held back during model training. Figure 4a shows that the relative $\mathrm{RMSE}_{P99}^{val}$ and $\mathrm{RMSE}_{P99}^{test}$ of the LSTM- and ConvLSTM models (colored circles and squares, respectively) lie close to the 1:1 line at all locations except Alicante (lightblue). This suggests that except at Alicante, the neural networks apply to unseen data relatively well in terms of their

error, which is further corroborated by the high correlations between the $\mathrm{RMSE}_{P99}^{val}$ and $\mathrm{RMSE}_{P99}^{test}$ across models at individual locations (Figure 4a). The MLR model and GTSM show similar behavior (colored triangles and diamonds, respectively). The $\mathrm{F1}_{P99}^{val}$ and $\mathrm{F1}_{P99}^{test}$ scores of all models also lie relatively close to the 1:1 line at most locations (Figure 4b), again suggesting that the models generalize reasonably well. However, the spread in the F1 scores is larger (as also seen in Figure 2) and the correlation between the $\mathrm{F1}_{P99}^{val}$ and $\mathrm{F1}_{P99}^{test}$ across the neural networks at each location is, although still visible and significant,

lower than between $\mathrm{RMSE}_{P99}^{val}$ and $\mathrm{RMSE}_{P99}^{test}$.

Moderate differences between $\mathrm{RMSE}_{P99}^{val}$ and $\mathrm{RMSE}_{P99}^{test}$, and $\mathrm{F1}_{P99}^{val}$ and $\mathrm{F1}_{P99}^{test}$, are expected because these error metrics depend on a relatively small number of extreme events that are not identically distributed in the relatively short validation



and test splits (<8 years each). This also affects how the performances of the different models compare. For instance, at Fishguard (pink) and Brest (grey), the minimum $\text{RMSE}_{P99}^{val}$ of the ConvLSTM models is slightly higher than that of GTSM,

while the minimum $\text{RMSE}_{P99}^{test}$ is slightly lower (Figure 4a). However, the discrepancy between $\text{RMSE}_{P99}^{val}$ and $\text{RMSE}_{P99}^{test}$ is exceptionally large at Alicante. Strikingly, this is the case for both the data-driven models and GTSM (Figure 4a). Given that the same atmospheric forcing was used for all models, this suggests that at Alicante, the extent to which the observed extremes can be explained by wind- and pressure-driven surges differs substantially between the validation and test splits. Potential reasons for this will be discussed in Section 4.

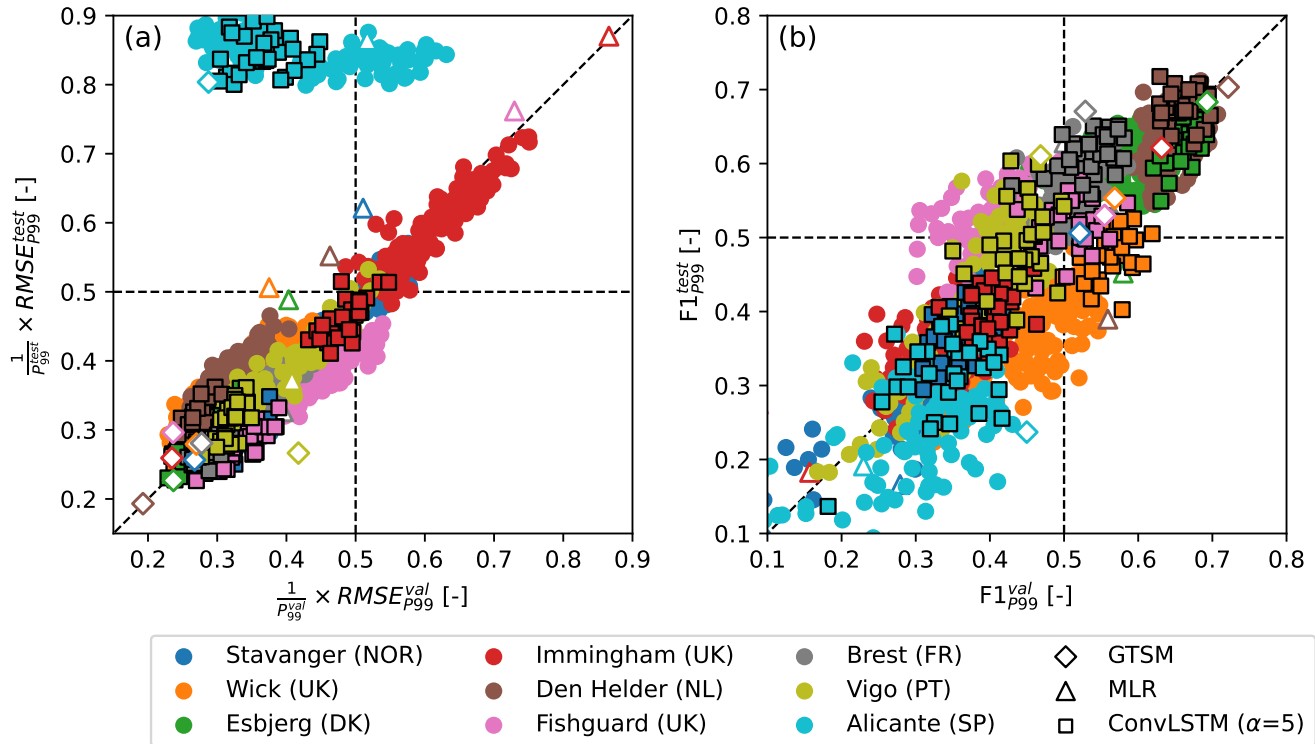

**Fig. 4.** Scatter plots of **(a)** $\text{RMSE}_{P99}$ relative to $P_{99}$ [-] in the validation split v.s. in the test split and **(b)** F1$_{P99}$ [-] in the validation split v.s. in the test split, for the LSTM models (circles), ConvLSTM models (black-edged squares), the MLR model of Tadesse et al. (2020) (triangles), and GTSM (Muis et al., 2020) (diamonds). Each marker denotes the error metrics of an individual model, and the colors represent the different tide-gauge locations. The diagonal lines indicate equal error metrics in the validation and test splits.





### 3.4 Underestimation of the highest extremes in the test split

To further investigate the performance of the neural networks at predicting extremes in the test split, we zoom in on the 6 locations at which the $\text{RMSE}_{P99}^{val}$ and $\text{F1}_{P99}^{val}$ of the ConvLSTM models and GTSM are comparable (Section 3.2) and the models generalize well (Section 3.3): Wick, Esbjerg, Den Helder, Fishguard, Brest and Vigo. We find that like in the validation split, the ConvLSTM models (orange) and GTSM (grey) have similar errors in the test split overall (average difference of 0.01 in the $\text{RMSE}_{P99}^{test}$ relative to $\text{P}_{99}^{test}$), whereas the LSTM models (blue) consistently have a higher error (Figure 5a). However, despite their similar $\text{RMSE}_{P99}^{test}$, the ConvLSTM models and GTSM have a different error distribution. Namely, the ConvLSTM models predominantly underestimate the observed extremes at all locations, while GTSM overestimates more than half of the observed extremes at Wick, Fishguard and Brest (Figure 5b).

The predictions from which the error distributions in Figure 5b were derived are shown in Figure C2. Because Figure C2 shows that the underestimation of extreme storm surges by the neural networks is more pronounced higher up the tail of the distributions of observed storm surges, we also computed the RMSE of predictions of exceedances of a higher threshold, namely $\text{P}_{99.9}^{test}$ (Figure 5c). Whereas the $\text{RMSE}_{P99}^{test}$ of the ConvLSTM models and GTSM are comparable, the $\text{RMSE}_{P99.9}^{test}$ of GTSM is lower than that of the ConvLSTM models at all locations (average difference of 0.07 in the $\text{RMSE}_{P99.9}^{test}$ relative to $\text{P}_{99.9}^{test}$). Comparing the error distributions in Figures 5b & d, we indeed find that the underestimation of extremes by the ConvLSTM models is more severe when using $\text{P}_{99.9}^{test}$ as a threshold. This contributes to a larger error compared to that of GTSM, which has predictions errors centered closer to 0 (Figure 5d). Nevertheless, the improvement by the ConvLSTM models relative to the LSTM models is still significant. Although these results are sensitive because the $\text{RMSE}_{P99.9}^{test}$ is based on only a small number of extremes, they suggest that the performance of the neural networks falls off in comparison to GTSM when considering extremes above very high percentiles.



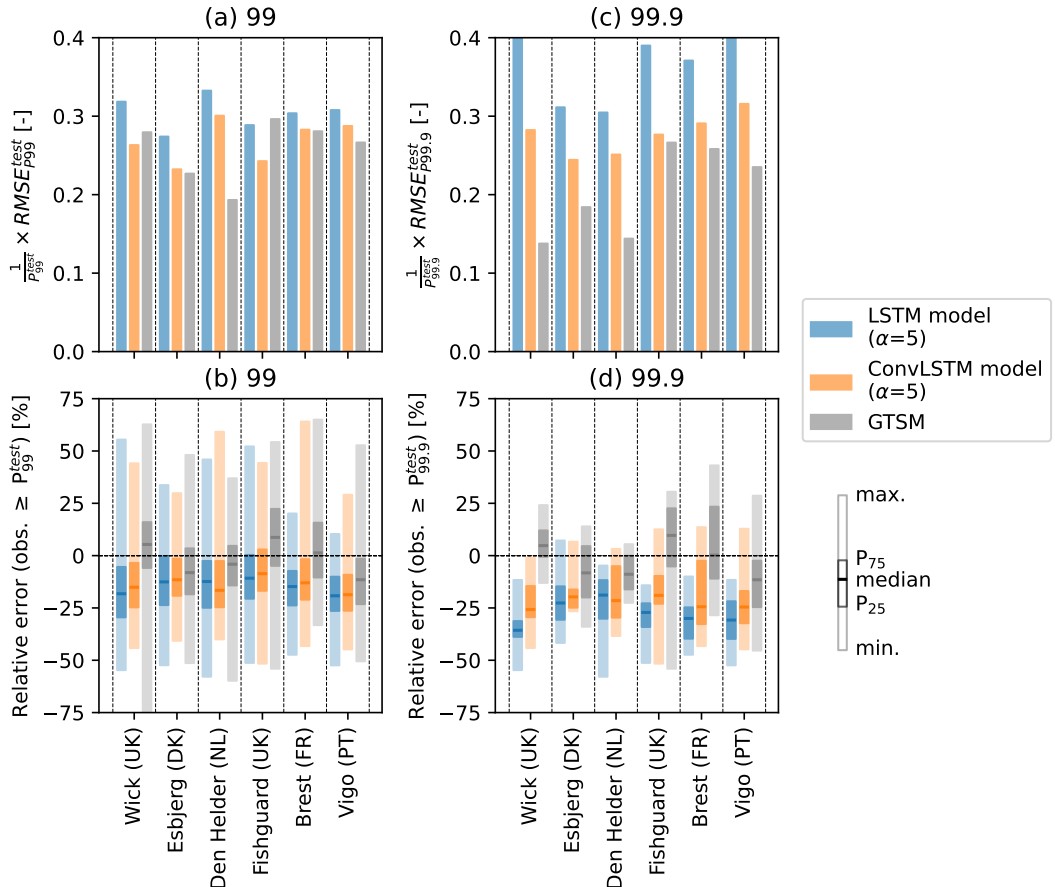

**Fig. 5. (a)** RMSE$_{P99}^{test}$ relative to P$_{P99}^{test}$, and **(b)** box plots of the distribution of the relative errors of the predictions of storm surges in the test split exceeding P$_{P99}^{test}$, at Wick (UK), Esbjerg (DK), Den Helder (NL), Fishguard (UK), Brest (FR) and Vigo (PT). Panels **(c)** and **(d)** show the same, but using P$_{P99.9}^{test}$ as a threshold for evaluating the prediction of observed extremes. Blue and orange colors are used to denote these metrics and distributions for the LSTM- and ConvLSTM models selected in Section 3.2 for $\alpha$=5, respectively, and grey colors for the hydrodynamic model GTSM.



**4 Discussion**

We found that training LSTM models with the density-based weighting devised by Steininger et al. (2021) improves both the error and timing of predicting extreme storm surge at 9 selected tide-gauge locations in Europe. This suggests that existing data-driven storm-surge models used for similar applications (e.g., Bruneau et al., 2020; Tiggeloven et al., 2021; Harter et al., 2024; Tadesse et al., 2020) could also be improved by addressing the underrepresentation of extremes in the training data in

this way. How much additional weight more extreme events should be given through the hyperparameter $\alpha$ depends on the location and the evaluation metrics to optimize (Section 3.1), and therefore needs to be tuned.

At most locations, using a ConvLSTM- instead of an LSTM model improves the predictions of the extreme storm surges. This conflicts with the results of Tiggeloven et al. (2021), who found that ConvLSTM models generally do not outperform LSTM models in Europe, nor globally. Most likely, the reason is that Tiggeloven et al. (2021) used atmospheric predictors in a

region of 1.25 by 1.25 degrees instead of 5 by 5 degrees around each tide-gauge location as a default. Given that extratropical cyclones occur at scales of hundreds of kilometers (Catto, 2016), more meaningful spatiotemporal features can likely be extracted from the predictor data when using a larger region. This is supported by a preliminary test at Esbjerg, which indicated that the LSTM- and ConvLSTM models indeed perform more similarly when trained with predictor data in a region of 1.25 by 1.25 instead of 5 by 5 degrees, and by sensitivity tests of Tiggeloven et al. (2021) with larger predictor regions.

Especially the ConvLSTM models perform relatively well at predicting extreme storm surges exceeding the 99th percentile, and their performance approximates that of the high-resolution, hydrodynamic model GTSM at the majority of locations (see Figures 3 & 5). This is promising, especially since GTSM was forced with ERA5 data at a higher frequency than the neural networks (see Section 2.1) and we did not tune hyperparameters other than the learning rate, the dropout rate, and $\alpha$ (see Section 2.3). Furthermore, depending on the application, a somewhat lower performance may be acceptable in exchange for the much

lower computational cost of applying the neural networks once trained. Follow-up research could therefore investigate the application of our neural networks to climate model simulations. This will introduce additional complexity because simulated distributions of predictor variables may differ from observed ones used for training due to climate-model biases and potential future changes (Lockwood et al., 2022). In this context, hydrodynamic model simulations forced with the same climate model simulations (e.g., Muis et al., 2023) could serve as a valuable benchmark.

At one location (Alicante), both the neural networks and GTSM performed reasonably in the validation split but poorly in the test split (Section 3.3), suggesting that the observed extremes in the test split can be explained by atmospherically-driven surges less well (see Section 3.3). Previous studies have also reported a lower performance of both data-driven and hydrodynamic models in southern Europe (e.g., Muis et al., 2020, 2023; Tadesse et al., 2020; Bruneau et al., 2020; Tiggeloven et al., 2021). A complicating factor is that storm surges in this area are small and the effect of other processes such as ocean

dynamic sea-level variability, freshwater forcing and waves are therefore relatively more important. Adding other predictors like temperature, precipitation, river discharge or waves, may help to represent these processes (Tadesse et al., 2020; Tiggeloven et al., 2021; Bruneau et al., 2020; Harter et al., 2024), but not all of these predictors are directly available from climate models.



We trained and evaluated the models using tide-gauge observations outside the harbor of Alicante because they are more complete, but tide-gauge observations inside the harbor are also available (Marcos et al., 2021; Haigh et al., 2021). Upon
comparison, we found that the two tide-gauge records have large differences in their extremes especially in the test split, and that both the predictions of the neural networks and the simulations of GTSM agree better with the observations inside the harbor. This signals the importance of waves, which affect the tide gauge inside the harbor less and are not (well) captured by the models. Another reason for the difference could be observational errors. To train the neural networks with less noisy data, hydrodynamic simulations could be used as the predictand instead of tide-gauge observations. The downside, however, is that
the neural networks will then inherit the biases of the hydrodynamic model and will not learn any indirect dependencies of observed extreme water levels on surface winds and sea-level pressure.

At two locations (Stavanger & Immingham), GTSM performed significantly better than the neural networks (Figure 3). Given that the models are forced by the same atmospheric data, this suggests that the neural networks at at least Stavanger and Immingham may be improved by further optimizing the neural networks. Since the optimal hyperparameters of neural
networks appear to be location-dependent (Tiggeloven et al., 2020), more extensive hyperparameter tuning than we did here may help to reduce both the performance differences between locations and between the neural networks and GTSM. This also includes the region of predictor data used at each tide-gauge location and the lag between the predictors and predictand, which we held constant.

Similarly to Harter et al. (2024), we find that the neural networks predominantly underestimate exceedances of very high
percentiles (e.g., the 99.9th), and perform worse than GTSM in that regard despite the density-based weights used for training (Section 3.4). Smaller errors may be obtained by increasing the density-based weights beyond the values that we tested, but likely at the cost of reduced precision (Section 3.1). Besides using more frequent predictor data, which would be unavailable from most climate models, follow-up research could explore several ways to reduce the underestimation.

First, the underestimation of the highest extreme storm surges partially occurs because the training data includes only few
similar events to learn from, requiring a higher degree of extrapolation by the neural networks. Therefore, it could be helpful to train the models with more data. This could be obtained from the backward extension of ERA5 to 1950 (Bell et al., 2021), depending on the length of the tide-gauge records. In the same spirit, the added value of complementing density-based weighting with synthetic oversampling of the extremes (e.g., Branco et al., 2017), and transfer learning across both different storm-surge datasets and different locations (e.g., Xu et al., 2023), would be useful to explore.

Second, follow-up research could explore whether the predictions of the extreme storm surges can be improved with advanced model architectures. For instance, path signatures, which encode features from time series through tensors of iterated path integrals, have shown promise as feature maps in machine learning tasks concerning irregular time series and the detections of extreme events (Riess et al., 2024; Lyons and McLeod, 2024; Akyildirim et al., 2022; Arrubarrena et al., 2024). Additionally, implementing self-attention mechanisms could help the neural networks to dynamically focus on those features
of the input data that are most relevant to the extremes (Ian et al., 2023; Wang et al., 2022). Another way to improve the accuracy may be to incorporate the shallow-water equations into the models, resulting in so-called physics-informed neural networks (e.g. Zhu et al., 2025; Donnelly et al., 2024).



## 5  Conclusions

We conclude that through density-based weighting, the cost-sensitive learning approach DenseLoss (Steininger et al., 2021)
improves the performance of neural networks at predicting extreme storm surges at all 9 selected tide-gauge locations in
Europe. Furthermore, at most locations, exploiting spatiotemporal dependencies using a ConvLSTM- instead of LSTM layer
also improves the performance, if a sufficiently large region of atmospheric predictor data is used. At 7 out of the 9 tide-gauge
locations that we used, the performance of especially the ConvLSTM models closely approximates that of the state-of-the-art,
hydrodynamic Global Tide and Surge Model (GTSM), based on performance metrics evaluated using the 99th percentile as
a threshold for extremes. This is a positive sign for the potential application of neural networks to climate model simulations
to project changes in extreme storm surges, especially since we trained the neural networks with 3-hourly data (the highest
frequency at which climate model simulations are typically provided) whereas GTSM was forced with hourly data. However,
the neural networks still predominantly underestimate the highest extreme storm surges (those exceeding the 99.9th percentile).
Follow-up research may improve this by further optimizing the neural networks and the data used to train them.

*Code availability.* The software that we developed to train and evaluate the models is publicly available on GitHub (https://github.com/
Timh37/surgeNN) and archived in a Zenodo repository (Hermans, 2025).

*Data availability.* We will publish a Zenodo repository with the data underlying this study upon final revision.

*Author contributions.* T.H.J.H. and C.B.H. conceptualized the study. T.H.J.H. developed the methodology, and analyzed and visualized the
results, with support of the other authors. T.H.J.H. wrote the manuscript. All authors contributed to reviewing and editing the manuscript.

*Competing interests.* The authors declare that they have no conflict of interest.

*Acknowledgements.* T.H.J.H. was supported by PROTECT. This project has received funding from the European Union's Horizon 2020
research and innovation programme under grant agreement no. 869304, PROTECT contribution number [TBD]. T.H.J.H. also received
funding from the NPP programme of NWO. J.J.M.B acknowledges funding from the National Science Foundation (Award 2019625). T.T.
received support from the MYRIAD-EU project, which received funding from the European Union's Horizon 2020 research and innovation
programme under grant agreement No 101003276. A.C. received support from the COMPASS project, which received funding from the
European Union's Horizon 2020 research and innovation programme under grant agreement No 101135481. We acknowledge the computing



and storage resources provided by the 'NSF Science and Technology Center (STC) Learning the Earth with Artificial intelligence and Physics (LEAP)' (Award #2019625).



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





## Appendix A: Neural network architectures

The LSTM model consists of an LSTM layer followed by 3 densely connected layers (Figure A1). For the LSTM layer, we
specified 32 units and otherwise used the default TensorFlow options. The first two densely connected layers have 32 neurons,
the commonly used rectified linear unit (ReLu) activation and L2 regularization ($l_2$=0.02), and are followed by a dropout layer
with a dropout rate that we lightly tuned (see Section 2.3). Regularization and normalization help to avoid overfitting the model
to the training data. The last dense layer has 1 neuron and a linear activation to predict a single storm surge at each time step.
The ConvLSTM model consists of a ConvLSTM instead of regular LSTM layer, with 32 kernels of 3 by 3 grid cells, even
padding and also a ReLu activation. The ConvLSTM layer is followed by batch normalization and a max-pooling layer that
reduces the spatial dimensions of identified features. The remainder of the ConvLSTM model is the same as in the LSTM
model.

For each prediction, predictors at time steps up to 24 hours prior were used (see Section 2.1), resulting in a total of nine
3-hourly time steps per prediction. The predictor data at each of the 20 by 20 grid cells and for each of the 4 predictor variables
shown in Figure 1 were stacked for the LSTM model, resulting in input data with the shape ($n_{obs}$,9,1600). Here, $n_{obs}$ refers
to the number of observations. For the ConvLSTM model, the grid cells were not stacked and the 4 predictor variables were
inputted as channels. The input to the ConvLSTM model therefore has the shape ($n_{obs}$,9,20,20,4).

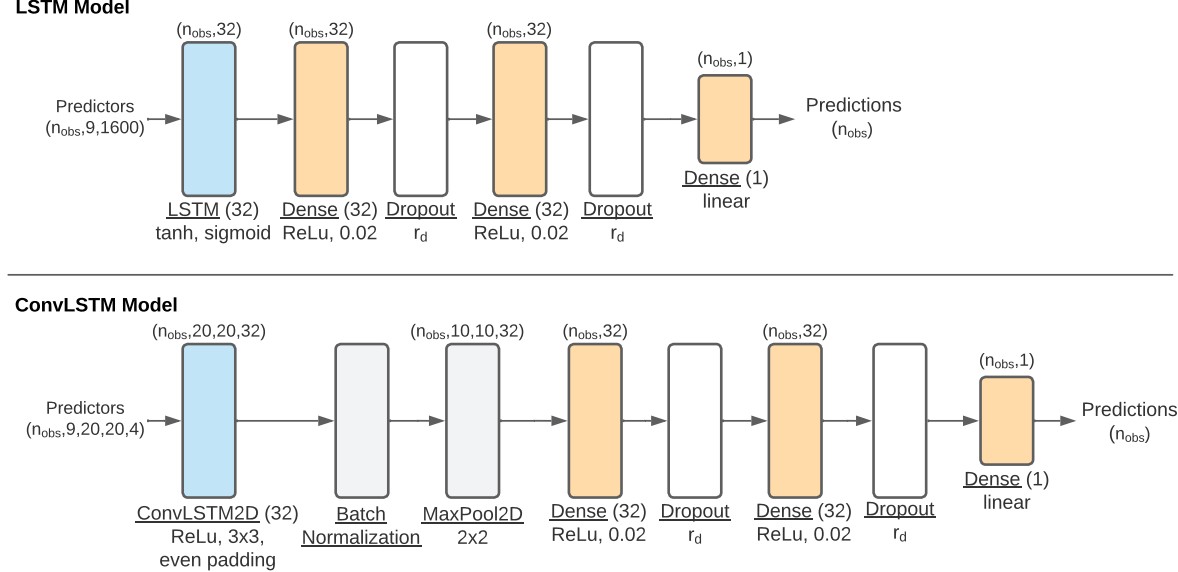

**Fig. A 1.** Flowchart of the architectures of the LSTM- and ConvLSTM models used. The blue rectangles represent the LSTM and ConvLSTM
layers, the orange rectangles the densely connected layers, the white rectangles the dropout layers and the grey layers the batch normalization
and max-pooling layers. The labels above the rectangles show how the shape of the data after passing through that layer. $r_d$ refers to the
tunable dropout rate.





## Appendix B: Density-based weights

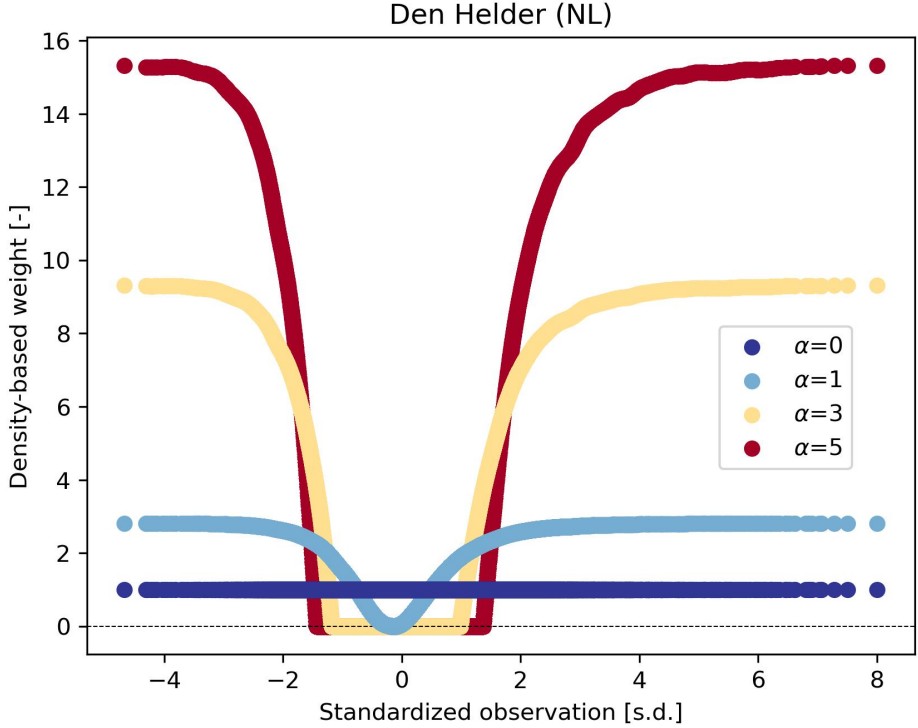

**Fig. B 1.** Density-based weights [-] of standardized observations [standard deviation (s.d.)] at Den Helder (NL) for $\alpha$ values of 0, 1, 3 and 5. Weights lower than $1\mathrm{e}^{-6}$ were clipped to $1\mathrm{e}^{-6}$ (see Section 2.3).





## Appendix C: Supplementary results





**Fig. C 1.** Scatter plots of the $\text{recall}_{P99}^{val}$ [-] versus the $\text{precision}_{P99}^{val}$ [-], for each tide-gauge location. Each circle denotes these error metrics for an individual LSTM model. The colors indicate the different values of $\alpha$ (0, 1, 3 or 5) used to train each LSTM model (30 LSTM models per $\alpha$ per location, as explained in Section 2.3). The bars on the bottom and left sides of each panel denote the minimum, median and maximum relative $\text{recall}_{P99}^{val}$ and $\text{precision}_{P99}^{val}$ of the LSTM models for each $\alpha$, respectively.



**Fig. C 2.** Scatter plots of predictions v.s. observed extremes ($\geq P_{99}^{test}$) in the test split [m] at **(a)** Wick (UK), **(b)** Esbjerg (DK), **(c)** Den Helder (NL), **(d)** Fishguard (UK), **(e)** Brest (FR) and **(f)** Vigo (PT). The blue and orange circles represent the predictions of the LSTM- and ConvLSTM models selected in Section 3.2 for $\alpha=5$, respectively. The grey circles represent the storm surges simulated with the hydrodynamic model GTSM. The vertical dashed line indicates the observed 99.9th percentile in the test split ($P_{99.9}^{test}$), and the diagonal 1:1 line denotes equal predictions and observations.