# Peer review of "Computing Extreme Storm Surges in Europe Using Neural Networks"

_EGUsphere, 2025_

## Author Comment (AC1)

**Reviewer 1**

The manuscript presents an approach to storm surge prediction using deep learning. Several methodological issues must be addressed to improve the manuscript's clarity and strength. Specifically, clarifying dataset construction, justifying hyperparameter choices, and improving performance evaluation will significantly enhance the manuscript. In its current form, the manuscript is not appropriate for publication.

We thank the reviewer for scrutinizing our methods. As further detailed below, we will address the reviewer's comments by better motivating the methodological choices that we made, providing additional details and discussion of aspects that were unclear before, and performing several tests to better justify the model parameters that we used. While we agree these revisions will strengthen our manuscript, our results and conclusions still apply.

Introduction:

L41-48: The last phrase "Furthermore, because several … hydrodynamics models" does not sound logical to me. It sounds like the authors reached a "general" conclusion from the previous "several" studies.

We will replace *"several"* by *"most"*, and restructure the sentence to "*Furthermore, how neural networks compare to state-of-the-art hydrodynamic models in this regard also remains unclear, because most previous studies either did not specifically evaluate the extremes or considered extremes exceeding relatively low thresholds (e.g., Bruneau et al., 2020; Tadesse et al., 2020; Tiggeloven et al., 2021)*" to improve its clarity.

Methodology – Data Preparation:

Dataset Size and Class Distribution: The paper mentions using data from 1979 to 2017 at a three-hour resolution. However, it is unclear how many training samples remain after filtering or how the extreme events (99% and 99.9%) are distributed.

Thank you for pointing this out. We will add a new table containing the total number of samples, the magnitude of the 99th and 99.9th percentiles, and the number of filtered extremes exceeding these percentiles for each tide gauge and split (Table B1, copied below). References to the new table will be added in Sections 2.4, 3.3 and 3.4. We will also clarify why the split sizes presented in Table B1 deviate slightly from the nominally defined split-size ratios in Section 2.3 by adding: *"Due to differences in tide-gauge data coverage between splits, the true split-size ratios can deviate from the nominal ones by up to a few percent (see Table B1)."* to L96.

**Table B1.** Number of samples, the magnitude of the 99th and 99.9th percentiles ($P_{99}$ and $P_{99.9}$) [m], and the number of filtered (see Section 2.4) extremes exceeding $P_{99}$ and $P_{99.9}$, per split and per tide gauge.

| Tide gauge | Samples [#] | | | $P_{99}^{split}$ [m] | | | $\geq P_{99}^{split}$ [#] | | | $P_{99.9}^{split}$ [m] | | | $\geq P_{99.9}^{split}$ [#] | | |
|---|---|---|---|---|---|---|---|---|---|---|---|---|---|---|---|
| | Train | Val | Test | Train | Val | Test | Train | Val | Test | Train | Val | Test | Train | Val | Test |
| 1 Stavanger | 65094 | 23178 | 21697 | 0.35 | 0.35 | 0.35 | 586 | 199 | 190 | 0.53 | 0.51 | 0.55 | 66 | 24 | 22 |
| 2 Wick | 57832 | 21538 | 16200 | 0.43 | 0.43 | 0.41 | 516 | 192 | 138 | 0.63 | 0.63 | 0.61 | 59 | 22 | 16 |
| 3 Esbjerg | 62889 | 20249 | 21364 | 1.07 | 1.09 | 1.10 | 572 | 187 | 192 | 1.85 | 1.69 | 1.72 | 63 | 21 | 22 |
| 4 Immingham | 52939 | 20280 | 20164 | 0.56 | 0.57 | 0.56 | 403 | 153 | 159 | 1.01 | 0.93 | 1.00 | 51 | 20 | 20 |
| 5 Den Helder | 66485 | 23368 | 23376 | 0.81 | 0.80 | 0.80 | 593 | 206 | 200 | 1.40 | 1.26 | 1.39 | 67 | 24 | 24 |
| 6 Fishguard | 57995 | 18361 | 19284 | 0.38 | 0.39 | 0.39 | 475 | 153 | 156 | 0.59 | 0.64 | 0.65 | 55 | 19 | 18 |
| 7 Brest | 67810 | 21748 | 23304 | 0.35 | 0.35 | 0.36 | 577 | 168 | 195 | 0.55 | 0.53 | 0.59 | 65 | 19 | 23 |
| 8 Vigo | 59772 | 22775 | 22663 | 0.29 | 0.29 | 0.30 | 487 | 168 | 207 | 0.45 | 0.42 | 0.45 | 53 | 16 | 23 |
| 9 Alicante | 53578 | 16692 | 19681 | 0.20 | 0.20 | 0.20 | 493 | 154 | 182 | 0.29 | 0.28 | 0.31 | 56 | 16 | 19 |

Explanatory Variables: While the paper includes zonal, meridional, and absolute wind speed as predictors, absolute wind speed is directly derivable from the other two. The authors need to justify this inclusion. Otherwise, removing absolute wind speed could prevent redundancy and improve efficiency.

We agree that absolute wind speed is a deterministic transformation of its zonal and meridional components. We initially included it based on prior work (Tiggeloven et al., 2021), who found that adding physically meaningful, deterministically derived predictor variables can sometimes improve model performance. Extensively testing the sensitivity of our models to the predictor variables was not our focus, but to still evaluate this we carried out an ablation study at one of the tide gauges (Esbjerg). This involved training the LSTM model with three different predictor sets: (1) only sea-level pressure, (2) sea-level pressure and zonal and meridional wind, and (3) sea-level pressure, zonal and meridional wind, and absolute wind speed (the default). We will include these experiments in a new appendix (Appendix A) and have copied the results below for the reviewer's convenience.

[Figure]

**Fig. A1 (top row).** Sensitivity of the average $RMSE_{P99}$ relative to $P_{99}$ [-] and $F1_{P99}$ [-] of the LSTM and ConvLSTM models at Esbjerg (DK) to the predictor variables (mean sea-level pressure psl, zonal and meridional wind u10 & v10, and absolute wind speed w10), for different values of α. The error metrics are shown for both the validation (1st and 3rd columns) and the test splits (2nd and 4th columns). The bold text on the left of the figure indicates the default settings used for the results in the main manuscript.

The results show that using the zonal and meridional wind components in addition to sea-level pressure clearly improves model performance, but that additionally using absolute wind speed does not. Importantly, however, using absolute wind speed does not clearly degrade overall model performance and generalization either. Based on these results, we conclude that absolute wind speed may indeed not be necessary, although we cannot exclude potential benefits at other locations. Since it did not degrade model performance at Esbjerg either, we therefore opt to keep absolute wind speed as a predictor variable at all sites. The detailed discussion of Figure A1 above will be added to the new Appendix A, and to clarify this in the manuscript, we will add the following:

"*Absolute wind speed was included based on previous research that showed that including derived but physically meaningful predictor variables can provide added value (Tiggeloven et al., 2021). Our sensitivity tests at the tide gauge in Esbjerg, however, suggest only a minor influence (see Appendix A). To improve model efficiency, future work could therefore investigate whether absolute wind speed could also be left out without substantially impacting model performance at other locations.*" to L83.

Construction of Data Points: Storm surges can persist for several days. It is essential to clarify whether data points overlap, whether each event is treated as an independent sample, or if multi-day storm surges are captured uniquely.
Threshold exceedances are treated independently to maximize the number of extreme samples available and allow the model to learn about the temporal evolution of storm surges. In other words, exceedances can be part of the same storm, and the predictor data used for them can partially overlap because of the 24h look-back window. As we agree these are important points to clarify, we will:

1) add *"Because of the look-back window, the predictor data used for predictions at consecutive 3-hourly time steps partially overlap."* at the end of L85.
2) add *"We treated remaining threshold exceedances independently regardless of whether they occurred during the same event, because this allows the neural networks to learn about the temporal evolution of storm surges, and uniquely capturing storm surges through declustering would reduce the available sample size unless more moderate events would be considered.The numbers of filtered exceedances in each split are shown in Table B1."* to L147.

Atmospheric Variables: The authors predict sea-level height based on ERA5 atmospheric data but do not specify whether land-based data is included. If land data is incorporated, the authors need to justify and discuss how it was handled.
As shown in Figure 1, the atmospheric predictor data includes values over land. We did not exclude this data because, as part of an atmospheric pattern over a given location that typically extends hundreds of kilometers, it may provide useful additional information for the prediction of storm surges. To clarify, we will add "*The predictor data includes grid cells over land, which do not directly affect water levels, but, as part of a certain weather pattern over a location, may contain features relevant for predicting storm surges.*" to L84.

Model Training & Hyperparameter Tuning:

Training Epochs: The authors use a maximum of 100 training epochs with early stopping. Given the complexity of the models, 100 epochs may not be sufficient. The authors need to justify the convergence of the model with 100 epochs.

We thank the reviewer for this comment as it prompted us to better motivate this choice. It was not feasible to visually inspect the loss evolution of all experiments, but we find that the maximum number of training epochs was reached in only 6.2% of all LSTM experiments and 5.6% of all ConvLSTM experiments, primarily when the lowest learning rate (1e-5) was used. In those instances, the average decrease in the validation loss over the last 5 decrements during training was only two to three tenths of a percent, with each decrement typically requiring multiple epochs. We therefore do not expect more training epochs to substantially improve our results.

We will add this justification to Section 2.3: *"The maximum number of epochs was reached for only 5 to 6% of all LSTM and ConvLSTM models, and based on the small decrements in the validation loss near the end of the training of these models, we do not expect that using a maximum of more than 100 training epochs would substantially improve our results."* (L130).

Dropout Rate: The dropout rate of 0.1–0.2 may be too low. LSTM and ConvLSTM models often use dropout rates of 0.3–0.5 to prevent overfitting. If different dropout rates have been tested, discussing their impact would improve transparency.

Upon comparison, we find that the agreement between the average performance metrics in the validation and test splits does not structurally improve with a dropout rate of 0.2 compared to a dropout rate of 0.1, suggesting that further increasing the dropout rate is not necessary to prevent overfitting and improve generalization in this case. To clarify, we will add the following note to Section 2.3 (L135): "*As a dropout rate of 0.2 did not lead to a structurally better generalization of the models to the independent test split than a dropout rate of 0.1, we did not increase the dropout rate beyond 0.2.*".

Performance Evaluation:

Evaluation Metrics: The authors use the F1-score as a primary evaluation metric. In extreme event prediction, recall is often more important than precision, as missing a storm surge event is more consequential than a false positive. A high F1-score does not necessarily indicate strong model performance if recall is low. Reporting recall and precision alongside the F1-score would provide a more comprehensive assessment. A confusion matrix could be also beneficial.

We agree that this may be the case in the context of flood forecasting, but for fitting extreme value statistics and projecting long-term changes in extremes, false positive predictions due to a lower precision would also adversely affect results, which is why we chose to use the F1-score alongside the RMSE. To better reflect this, we will change L199-200 *"Namely, increasing the density-based weights generally leads to more true positives but also to more false positives"* to "*Namely, increasing the density-based weights generally leads to a higher recall but a lower precision (i.e., less false negatives but also more false positives)*" and explicitly mention that the decreasing precision of neural networks trained with a higher $\alpha$ parameter may "*negatively influence subsequent extreme-value analyses.*" (L194). We

agree these aspects are important, which is why the recall and precision as a function of $\alpha$ are shown for the different locations in Fig. C1 (will become Fig. D2), which is referred to in Section 3.1. To acknowledge that in some contexts, recall may be more important than precision, we will add to the discussion in Section 4 that the hyperparameter $\alpha$ needs to be tuned *"in relation to the problem context. For instance, a higher $\alpha$ value may be better for applications in which recall is more important than precision (see Fig. D2)."* (L281).

Discussion & Conclusion:

The discussion and conclusion are well-written based on the current results of the study, but they will need to be updated after the revision of the manuscript.
We thank the reviewer for their positive feedback on our discussion and conclusions. Our revisions in response to the reviewer's comments will mainly involve improving the explanation and justification of our methods. As detailed above, major updates to the discussion and conclusion sections were not necessary.

Minor Edits:

"v.s." to "vs."
This will be corrected throughout the paper.

L41: "more moderate" to "moderate."
Thank you. We will remove *"more"*.

L318: Remove "at least."
We will replace *"at least"* with *"especially"*.

**Additional changes (to be appended to both responses)**
  1. We discovered a minor error in the stratification routine introduced in Section 2.1, where 1 year of input data was accidentally omitted at the locations Wick and Alicante. We fixed this error, trained the models again at these locations and will update the corresponding results throughout the paper where needed. Fixing the error led to only minor differences in the results.
  2. Figure C1 (now D2) accidentally showed results for the test split, instead of for the validation split. We apologize for this mistake, and will replace the figure with its correct version (see below). As the general effect of α on precision and recall is consistent across the validation and test splits, the discussion of Figure C1 (now D2) in Section 3.1 will not need to be changed.

[Figure]

Corrected version of Fig. C1

3. Figure 4 aggregates results for all locations, which we realized makes it harder to inspect model generalization at individual locations. We will therefore complement Figure 4 with an additional supplementary figure (Figure D3) that shows the same results, but separated by location (see below).

[Figure]

**Fig. D 3.** Scatter plots of RMSE$_{P99}$ relative to P$_{99}$ [-] and F1$_{P99}$ [-] in the validation vs. in the test split, displayed per tide gauge. The colored circles represent the LSTM models for different values of α, the black-edged red squares the ConvLSTM models for α=5, and the

white triangles and diamonds the MLR model of Tadesse et al. (2020), and GTSM (Muis et al., 2020), respectively. The diagonal line in each panel indicates equal error metrics in the validation and test splits.

A few lines of discussion of Figure D3 will be added to Section 3.3: "*Additionally, we find that increasing α leads to a lower $RMSE_{P99}$ in both the validation and test splits (Figure D3).*" (L241), "*With a few exceptions, increasing α has an approximately similar effect on $F1^{test}_{P99}$ as it has on $F1^{val}_{P99}$ (Figure D3).*" (L245), and "*For optimal model generalization, we therefore recommend tuning α alongside other important (hyper)parameters using k-fold cross-validation.*" (L248).

---

## Author Comment (AC2)

**Reviewer 2**
This paper presents an investigation into the use of neural networks (NNs) for predicting extreme storm surges. The core contribution is the application and evaluation of a cost-sensitive learning approach (DenseLoss) to specifically improve the prediction of the rare, high-impact events. Two NN architectures (LSTM and ConvLSTM) were compared against both a simpler statistical model (MLR) and a hydrodynamic model (GTSM) across nine European tide-gauge locations. This is a well-written paper on an interesting topic. I believe the manuscript could be strengthened by considering the following points.
We thank the reviewer for their positive feedback and the opportunity to strengthen our manuscript.

The paper identifies data imbalance as a major issue, but the choice of DenseLoss needs stronger justification. It's basically a simple re-weighting technique—and more advanced options (like SMOGN) can be used?
We agree that the underrepresentation of extreme events in regression tasks form a nontrivial challenge, which we emphasize has not yet been addressed in the context of predicting extreme storm surges. Indeed, several strategies have been proposed to handle data imbalance, including data-level approaches like SMOGN (Branco et al., 2017). In our study, we deliberately opted for DenseLoss (Steininger et al., 2021) based on the following considerations:

- Unlike resampling strategies, cost-sensitive approaches like DenseLoss operate at the algorithm level and do not require altering the training dataset, thereby preserving the physical structure of the input data and avoiding degradation of generalization due to the addition of redundant or noisy samples. We feel the former is an especially relevant drawback in our case as interpolating between high-dimensional and structured spatiotemporal atmospheric data (9 time steps, 20 by 20 grid cells, 4 variables) to generate new synthetic events is not trivial and may violate physical consistency.
- The DenseLoss method is straightforward to implement and scale in deep learning pipelines while maintaining a consistent and stable training process across different locations. It includes a single interpretable hyperparameter ($\alpha$) that allows users to control the emphasis on rare samples via kernel density estimation on the observed target distribution. In contrast, implementing SMOGN would require the definition of a (location-dependent) relevance function and thresholds for binning the original training data, and configuring complex feature interpolation schemes, particularly given the nature of our predictor data.
- Although applied in a different setting, Steininger et al. (2021) showed that DenseLoss consistently outperforms SMOGN across both synthetic and real-world datasets, while requiring fewer assumptions about the distribution of the training data.

To better justify our choice for DenseLoss in the manuscript, we will add a summary of these considerations to Section 2.3:

"*Therefore, we implemented the cost-sensitive learning approach DenseLoss (Steininger et al., 2021), which is an algorithm-level method that reweights the loss function based on the rarity of target values. In contrast to resampling methods such as Synthetic Minority*

*Oversampling with Gaussian Noise (SMOGN; Branco et al., 2017), DenseLoss does not alter the training data through synthetic oversampling, and therefore retains the physical consistency of the high-dimensional training data that we use. Furthermore, the additional emphasis placed on rare samples can be controlled through a single interpretable hyperparameter and does not require an a-priori relevance definition.*" (L119), and a reference to the benchmarks of Steiniger et al. (2021) at the end of the paragraph.

Does just up-weighting rare events actually help the model learn their complex, nonlinear physics, or does it merely force better scores on a few outliers while impacting overall physical consistency?

We argue that the DenseLoss method does not just force better scores on a few outliers because: (i) the introduction of density-based weights does not only improve the RMSE of the predictions of observed exceedances of the 99th percentile, but also the F1 score. This indicates that rather than reducing the prediction error of a few specific outliers, DenseLoss contributes to a better characterization of the extreme events in general; (ii) the very high extremes are still predominantly underestimated (Fig. 5d), which further supports that the models are not overcompensating by forcing better scores on those few highest-weighted target outliers.

To incorporate this in the manuscript, we will add: *"Our finding that using DenseLoss can improve both $RMSE_{val}^{P99}$ and $F1_{val}^{P99}$ suggests that reweighting the loss function does not simply reduce prediction errors of a few specific outliers but improves to the models' overall representation of extremes"* to L196 and *"While this indicates that by density-based weighting, the neural networks are not overcompensating by forcing good scores on only a few high-weighted outliers,"* to L325.

The conclusions are based on an experimental setup with several fixed, important parameters—like using just nine tide gauges. Why these, and do they really capture Europe's varied coastal dynamics?

The tide gauges were chosen based on their broad geographical coverage and the available tide gauge data during 1979-2017. The number of tide gauges was limited due to computational constraints. While we agree their locations may not exhaustively represent all European coasts, these tide gauges are diverse in terms of their shoreline orientation, tidal regime, locally important dynamics and the magnitude and distribution of observed extremes. These differences are reflected by the location-dependence of some of our results. We will clarify this rationale by elaborating L75-L77: *"Due to computational constraints, we limited our experiments to 9 tide gauges. While these may not be representative of all European coasts, they allow us to compare results across locations that are diverse in terms of shoreline orientation, dynamics, tidal regime and the magnitude and distribution of extremes."*.

And how did you choose a 5×5° domain and a 24-hour lookback? A sensitivity analysis would show whether your results hold up when these parameters change.

A 5° x 5° domain was chosen as a compromise between computational costs and the approximate length scales at which we expect remote winds and pressure to affect local water levels. A 24-hour lookback was chosen based on the approximate 48-hour duration of a storm surge, which under the assumption of a symmetrical hydrograph, implies that atmospheric information in the 24 hours preceding extreme water levels is most relevant.

Both these motivations will be added to the second paragraph of Section 2.1: "*This domain size was chosen as a compromise between computational costs and the approximate spatial scales at which we expect remote winds and sea-level pressure to be relevant*" and "*which was based on the assumption of a typical storm surge duration of approximately 48 hours.*", respectively.

Given the aims of our study, we kept the domain and lookback window constant across the different locations, but we agree with the reviewer that gaining more insight into the sensitivity of the models to these parameters would be useful. Therefore, we carried out new sensitivity tests at one of the tide gauges (Esbjerg), varying the length of the lookback window (between 0, 12, 24 and 36 hours) and the predictor domain size (between 1 by 1, 3 by 3 and 5 by 5 degrees). We will include these experiments in a new appendix (Appendix A) and have copied the results below for the reviewer's convenience.

[Figure]

**Fig. A1 (bottom three rows).** Sensitivity of the average $RMSE_{P99}$ relative to $P_{99}$ [-] and $F1_{P99}$ [-] of the LSTM and ConvLSTM models at Esbjerg (DK) to length of the look-back window (0, 12, 24 or 36 hours), and the domain size of the predictor data (1 by 1, 3 by 3 or 5 by 5 degrees), for different values of α. The error metrics are shown for both the validation (1st and 3rd columns) and the test splits (2nd and 4th columns). The bold text on the left of the figure indicates the default settings used for the results in the main manuscript.

The new sensitivity tests indicate that the LSTM models trained with a predictor region of 3 by 3 or 5 by 5 degrees tend to outperform LSTM models trained with a domain size of 1 by 1 degrees, but not by much. The ConvLSTM models more clearly benefit from a larger domain

size than the LSTM models, and as a result, they outperform the LSTM models when using a predictor region of 3 by 3 or 5 by 5 degrees (as discussed in Section 4).

Secondly, we find that using a look-back window for the predictor data is clearly better for the performance of the LSTM models than using no look-back window, which is what several previous studies did. A look-back window of 24 hours seems to be approximately optimal, and increasing the look-back window to 36 hours does not further improve the performance of the models, at least not in this case.

Although we did not test different combinations of these settings and the optimal configuration of the predictor data may vary by location, the new sensitivity tests suggest our parameter choices are generally appropriate. Follow-up research could investigate fine-tuning the predictor data further. We will add this detailed discussion to Appendix A. Additionally, we will incorporate the headlines of these tests in the main manuscript by adding: "*Sensitivity tests at Esbjerg suggest that these parameter choices are generally appropriate (see Appendix A), although we acknowledge that the optimal configuration of the predictor data may vary by location. As shown in Appendix A, using a look-back window, which several previous studies did not do (Bruneau et al., 2020; Tiggeloven et al., 2021; Harter et al.,2024), clearly provides added value.*" to L85 and rewriting L322 to say that tuning the models more extensively, as proposed in Section 4, *"could also involve optimizing the variables, domain size and look-back window of the predictor data used at each tide-gauge location (see Appendix A).*".

Based on your results, the NNs still tend to underestimate the very highest extremes (99.9 percentile). Since accurate tail behavior is key for hazard assessment, it might help to investigate why this happens. Is it a smoothing effect in the ERA5 reanalysis, or a limit in the network's ability to extrapolate even with DenseLoss? A brief investigation could really strengthen your conclusions.

We agree that accurately predicting storm surges in the upper tail is important. Our study forms a step toward that goal by addressing the underrepresentation of extremes in observations, which we emphasize is something that previous studies did not consider (e.g., Bruneau et al., 2020; Tiggeloven et al., 2021; Harter et al., 2024). Given this, we find that our results regarding the predominant underestimation of the exceedances of the 99.9th percentile despite density-based weighting and the performance of the neural networks relative to GTSM are both relevant and timely. As already discussed in Section 4, the predominant underestimation has different potential reasons, related to (1) limitations of the LSTM and ConvLSTM architectures, (2) the tuning of the models, (3) the training data used (and the temporal frequency of the ERA5 data in particular). In our view, disentangling these reasons would require dedicated follow-up research. To be more complete and explicit about potential follow-up studies that could shed more light on these aspects, we will make the following changes:

- We will add: *"Another reason for the predominant underestimation of exceedances of the 99.9th percentile could be the limited ability of the neural networks to extrapolate to the highest extremes, despite our use of DenseLoss.*" at the end of L325 to more clearly introduce the last two paragraphs discussing relevant follow-up research to address this.

- We will add: *"more extensive hyperparameter tuning than we did here may help to reduce both the performance differences between locations and between the neural networks and GTSM, including at predicting the highest extremes."* (L321) to emphasize the potential for further optimizing the models that we use regarding the underestimation of the highest extremes.

Furthermore, as we previously referred to only briefly at the end of Section 2.4, Agulles et al. (2024) compared numerical ocean model simulations with either hourly or daily ERA5 forcing. They found that the latter leads to approximately 30-50% lower estimates of the 99.9th percentile of non-tidal residuals along European coastlines. Although the differences between hourly and 3-hourly forcing would likely be smaller and would need to be tested to draw definitive conclusions, the results of Agulles et al. (2024) suggest that the temporal frequency of the ERA5 data that we used does contribute to the underestimation of the observed exceedances of the 99.9th percentile by our data-driven models, and provide a rough quantitative idea of the upper limit on how much. We will therefore also add the following to the discussion (L326):

*"Agulles et al. (2024) found that by using daily instead of hourly atmospheric forcing, their hydrodynamic model underestimated the 99.9th percentile of non-tidal residuals by approximately 30-50%. While the underestimation with 3-hourly instead of hourly forcing would likely be less severe, their results suggest that the difference in the temporal frequency of the forcing of the neural networks and GTSM explains the differences in their tail behaviour at least partially. Future work could investigate this further by running a hydrodynamic model with 3-hourly forcing."*

Noting that the best α parameter changes from site to site raises practical challenges: is the model capturing general patterns or simply fitting each location's unique data distribution? If you must tune α for every gauge, rolling this out to hundreds of sites becomes both computationally heavy and methodologically challenging.

We thank the reviewer for highlighting the practical implications of our finding that the optimal α value for reweighting depends on the location, and agree that introducing an additional parameter to tune at each location makes large-scale deployment more challenging. At the same time, we emphasize that our finding intuitively makes sense because the (tails of the) distribution of observed water levels varies by location. Nevertheless, our results in Fig. 2 suggest that even for a diverse set of locations, a common value for α>0 can be found that improves the performance of the data-driven models at predicting extremes at all including locations, even though it may not lead to the most optimal performance at every location individually.

Tuning α for a large number of locations could be automated using meta-learning or Bayesian optimization techniques (e.g., Feurer and Hutter, 2019). Alternatively, the optimal weighting could be estimated from distributional features of the training data — such as target skewness or tail heaviness — using learned heuristics or data-driven priors. More practically, we envision clustering locations based on the similarity of the characteristics and distribution of their extremes (e.g., Calafat & Marcos, 2020; Rashid et al., 2024), and tuning the DenseLoss method for a reasonable number of such clusters. This could be done either by training separate models for each cluster of locations or by training multi-site models similar to the one of Kratzert et al. (2019) and incorporating cluster-specific loss weights.

As we agree with the reviewer this is an important point, we will add the following paragraph to the discussion: "*The optimal α value for reweighting the loss function value likely varies by location because the distribution of the training data also varies by location. To apply the DenseLoss method to a larger number of locations, the location-dependent tuning of α could be automated (e.g., Feurer and Hutter, 2019) or informed by the distributional features of the training data, such as target skewness and tail heaviness. More practically, α could be tuned for a limited number of clusters of locations with similar characteristics and distributions (e.g., Calafat and Marcos, 2020; Rashid et al., 2024). This could either be done by training cluster-specific models or by training a single model incorporating cluster-specific attributes and weights (e.g., Kratzert et al., 2019). Our results in Figure 2 suggest that even for a diverse set of locations, a common value for α>0 can be found that improves the performance of the data-driven models at predicting extremes at all locations, even though it may not lead to the most optimal performance at every location individually.*" (L282).

Feurer, M. and Hutter, F., 2019. Hyperparameter optimization. In *Automated machine learning: Methods, systems, challenges* (pp. 3-33). Cham: Springer International Publishing.
F.M. Calafat, & M. Marcos, Probabilistic reanalysis of storm surge extremes in Europe, Proc. Natl. Acad. Sci. U.S.A. 117 (4) 1877-1883, https://doi.org/10.1073/pnas.1913049117 (2020).
Morim, J., Wahl, T., Rasmussen, D.J. et al. Observations reveal changing coastal storm extremes around the United States. Nat. Clim. Chang. 15, 538–545 (2025). https://doi.org/10.1038/s41558-025-02315-z
Kratzert, F., Klotz, D., Shalev, G., Klambauer, G., Hochreiter, S. and Nearing, G., 2019. Towards learning universal, regional, and local hydrological behaviors via machine learning applied to large-sample datasets. *Hydrology and Earth System Sciences*, *23*(12), pp.5089-5110.

Perhaps the comparison would be more balanced if both models used the same input cadence (ConvLSTM and GTSM). The ConvLSTM is driven by 3-hourly data, whereas GTSM benefits from hourly forcing, which may contribute to its sharper extreme peaks. Ideally, the authors could run GTSM on the same 3-hourly inputs; if that isn't feasible, a clearer justification for the differing cadences would be helpful.

We agree with the reviewer that the comparison likely favors GTSM because of its higher forcing frequency and ideally GTSM simulations with 3-hourly ERA5 forcing would be used. Unfortunately, Muis et al. (2020 & 2023) only ran GTSM with hourly forcing and we do not have the access nor capacity to run such simulations ourselves. We nevertheless used the GTSM simulation because it still forms a valuable benchmark for our data-driven models: where the performance of the data-driven models trained with 3-hourly forcing is close to that of GTSM forced with hourly data, the data-driven models clearly perform well, and where the data-driven models perform substantially worse compared to GTSM forced with hourly data, they would likely also perform worse than GTSM forced with 3-hourly data. To better explain why we use differing cadences in the manuscript, we will rewrite L185-188 to read: *"A limitation of the comparison with GTSM is that we trained the data-driven models with 3-hourly ERA5 data (see Section 2.1), but simulations of GTSM are only available forced with hourly instead of 3-hourly ERA5 data. As atmospheric forcing with a lower temporal resolution reduces the accuracy of storm surge models (Agulles et al., 2024), our comparison is biased towards GTSM in this regard. We will consider this for the interpretation of our results in the following sections."*

The paper would be improved by a brief discussion of its findings in the context of other advanced architectures. The authors should consider contextualizing their work with respect to models like Graph Neural Networks (GNNs), hierarchical deep neural networks, and Gaussian Process models, which have been successfully applied to similar spatiotemporal problems. This would provide valuable perspective on why LSTM/ConvLSTM were chosen and how they fit within the rapidly evolving field.

Our primary objective was to build upon previous studies by studying cost-sensitive learning as a means to improve predictions of extreme storm surges, rather than extensively testing different machine-learning models. As described in Section 2.2, we therefore chose the models that were identified to perform best by a comparable previous study (Tiggeloven et al., 2021). Additionally, we argue that these architectures are appropriate for our data as they can capture temporal dependencies and local spatial structure, without requiring the explicit graph design of GNNs or the complexity of a hierarchical model. We will make this rationale more explicit by rephrasing L113-L115.

Recent advances in the field may indeed help to improve the prediction of storm surges. In the last paragraph of the discussion, we already highlighted several potential avenues for investigating alternative architectures, such as path signatures, attention mechanisms and physics-informed neural networks. As the models put forward by the reviewer are also promising for the prediction of storm surges, we agree that briefly discussing them is worthwhile. We will therefore expand the existing discussion by adding: *"Second, while we chose to use LSTMs and ConvLSTMs (see Section 2.2), follow-up research could investigate whether the predictions of the extreme storm surges can be improved with other, emerging model architectures. For instance, graph neural networks, hierarchical deep neural networks and gaussian process models have been found beneficial for short-term forecasting (Kyprioti et al., 2023; Jiang et al., 2024; Naeini et al., 2025), and may also be in our context. Graph neural networks in particular could help to predict storm surges at multiple related locations, capturing spatial dependencies by representing different locations as nodes of a graph."*

Minor Comment:

Line 487 (Appendix A): "Regularization and normalization help to avoid overfitting..." It should be "Regularization and dropout help to avoid overfitting...". Batch normalization serves a different primary purpose (stabilizing and accelerating training).

Thank you for pointing this out. We will replace '*normalization*' with '*dropout*'.

**Additional changes (to be appended to both responses)**
1. We discovered a minor error in the stratification routine introduced in Section 2.1, where 1 year of input data was accidentally omitted at the locations Wick and Alicante. We fixed this error, trained the models again at these locations and will update the corresponding results throughout the paper where needed. Fixing the error led to only minor differences in the results.
2. Figure C1 (now D2) accidentally showed results for the test split, instead of for the validation split. We apologize for this mistake, and will replace the figure with its correct version (see below). As the general effect of α on precision and recall is consistent across the validation and test splits, the discussion of Figure C1 (now D2)

[Figure]

**Corrected version of Fig. C1**

[Figure]

3. Figure 4 aggregates results for all locations, which we realized makes it harder to inspect model generalization at individual locations. We will therefore complement Figure 4 with an additional supplementary figure (Figure D3) that shows the same results, but separated by location (see below).

**Fig. D 3.** Scatter plots of RMSE$_{P99}$ relative to P$_{99}$ [-] and F1$_{P99}$ [-] in the validation vs. in the test split, displayed per tide gauge. The colored circles represent the LSTM models for different values of α, the black-edged red squares the ConvLSTM models for α=5, and the

white triangles and diamonds the MLR model of Tadesse et al. (2020), and GTSM (Muis et al., 2020), respectively. The diagonal line in each panel indicates equal error metrics in the validation and test splits.

A few lines of discussion of Figure D3 will be added to Section 3.3: "*Additionally, we find that increasing α leads to a lower $RMSE_{P99}$ in both the validation and test splits (Figure D3).*" (L241), "*With a few exceptions, increasing α has an approximately similar effect on $F1^{test}_{P99}$ as it has on $F1^{val}_{P99}$ (Figure D3).*" (L245), and "*For optimal model generalization, we therefore recommend tuning α alongside other important (hyper)parameters using k-fold cross-validation.*" (L248).